# Perinatal Outcomes of Diet Therapy in Gestational Diabetes Mellitus Diagnosed before 24 Gestational Weeks

**DOI:** 10.3390/nu16111553

**Published:** 2024-05-21

**Authors:** Yoshifumi Kasuga, Marina Takahashi, Kaoru Kajikawa, Keisuke Akita, Toshimitsu Otani, Satoru Ikenoue, Mamoru Tanaka

**Affiliations:** Department of Obstetrics and Gynecology, Keio University School of Medicine, 5 Shinanomachi, Shinjuku-ku, Tokyo 160-8582, Japansikenoue.a3@keio.jp (S.I.); mtanaka@keio.jp (M.T.)

**Keywords:** gestational diabetes mellitus, diet therapy, insulin, body mass index, gestational weight gain, oral glucose tolerance, large for gestational age, preconception care

## Abstract

To evaluate perinatal outcomes and risk factors for large for gestational age (LGA; birth weight over 90 percentile) in gestational diabetes diagnosed before 24 gestational weeks and treated with diet therapy alone until delivery (Diet Early gestational diabetes mellitus (Diet Early GDM)), we assessed the maternal characteristics and perinatal outcomes of patients with early GDM (*n* = 309) and normal glucose tolerance (NGT; *n* = 309) at Keio University Hospital. The gestational weight gain (GWG) expected at 40 weeks was significantly lower in the Diet Early GDM group than in the NGT group. The Diet Early GDM group exhibited a significantly lower incidence of low birth weight (<2500 g) and higher Apgar score at 5 min than the NGT group. Multiple logistic regression analysis revealed that the pre-pregnancy body mass index and GWG expected at 40 weeks were significantly associated with LGA for Diet Early GDM. No differences were observed in random plasma glucose levels in the first trimester, 75 g oral glucose tolerance test values, and initial increase or subsequent decrease between the two groups. Dietary early GDM did not exhibit a worse prognosis than NGT. To prevent LGA, it might be important to control maternal body weight not only during pregnancy but also before conception.

## 1. Introduction

Gestational diabetes mellitus (GDM) is a high-incidence perinatal complication, and neonates born to mothers with GDM are at higher risk of large for gestational age (LGA: birthweight ≥ 90% percentile), macrosomia (birthweight ≥ 4000 g), neonatal hypoglycemia, and jaundice [1]. GDM is a prenatal metabolic disease that is treated with diet therapy, exercise, oral medications (e.g., sulfonylureas and metformin), and insulin. The first step in managing GDM is to recommend lifestyle changes, including nutrition and exercise, to improve hyperglycemia [2]. Nutritional management is considered fundamental for GDM management [3]. Several nutrition management guidelines recommend a caloric intake between 1500 and 2000 kcal/day, and a reduced calorie intake is recommended for overweight (pre-pregnancy body mass index [BMI]: 25 ≤ BMI < 30 kg/m^2^) and obese (pre-pregnancy BMI ≥ 30 kg/m^2^) mothers with GDM [4]. However, reduced calorie intake as a dietary therapy for mothers with GDM to improve short- and long-term prognoses remains unclear. Since diet therapy is the bedrock, in GDM that is diagnosed after 24 gestational weeks (Late GDM), the diet therapy group is usually considered to be a lower risk group for perinatal outcomes than the insulin therapy group. However, a Swedish population-based cohort study reported that the diet therapy group with Late GDM should not be managed as a low-risk group because it did not normalize the perinatal prognosis [5]. Therefore, patients with Late GDM treated with diet therapy alone should not be managed as a low-risk group.

Recently, GDM diagnosed in the second trimester has become the focus of perinatal management. In 2023, the Treatment of Booking Gestational Diabetes Mellitus (TOBOGM) trial, which evaluated the neonatal outcomes of GDM diagnosed before 20 gestational weeks, reported that adverse neonatal outcomes in the immediate treatment group were lower than those in the non-immediate treatment group [6]. In Japan, GDM is diagnosed before 24 weeks of gestation (Early GDM) and treated. Early GDM was diagnosed using the criteria published by the Japanese Society of Obstetrics and Gynecology and our hospital’s criteria described in our previous reports [7,8]. The pre-pregnancy BMI and incidence of a family history of diabetes in the Early GDM group were significantly higher than those in the Late GDM group [7]. Furthermore, the incidence of LGA and macrosomia in Early GDM was significantly higher than that in Late GDM [9,10]. Therefore, Early GDM is considered a higher-risk group than Late GDM. Because neonates with LGA and macrosomia have a higher risk of future obesity and metabolic syndrome [11], fetal overgrowth should be prevented. Diet therapy for GDM is required to ensure appropriate fetal growth as a consequence of appropriate maternal weight gain during pregnancy [3]. However, for Early GDM, suitable diet therapy for perinatal outcomes remains unknown, and perinatal outcomes of diet therapy have not been evaluated in Early GDM.

Therefore, we aimed to investigate perinatal outcomes and risk factors for LGA in patients with early GDM treated with diet therapy alone until delivery (Diet Early GDM).

## 2. Materials and Methods

This was a retrospective cohort study. Of the singleton expectant women who received perinatal care at Keio University Hospital between January 2013 and December 2021, 313 patients with early GDM who were treated with diet therapy alone until delivery and had normal glucose tolerance (NGT) were included in the study. The diagnostic criteria for GDM in this study are shown in Appendix A [7,8]. NGT was defined as expectant women who had a normal 50 g glucose challenge test (GCT), or abnormal 50 g GCT but a normal 75 g oral glucose tolerance test (OGTT), at 24–28 gestational weeks, and they were recruited from the perinatal database at our hospital during the same study period (*n* = 3269) by propensity score matching (PSM). The score was estimated using a logistic regression model and greedy matching (ratio of 1:1 and matching without replacement) with a caliper width and standard deviation of 0.20. The factors used for PSM were maternal age, parity, and pre-pregnancy BMI. Finally, the balance of each covariate between the early GDM and NGT groups was evaluated using standardized differences (between-group difference/pooled standard deviation). An absolute standardized difference value of <10% was considered a relatively small imbalance [12]. All study participants provided informed consent.

Management of Early GDM has been described in our previous reports. All expectant mothers with early GDM received diet therapy from nutritionists, including three meals and snacks. Capillary glucose profiles were obtained seven times per day under dietary management via self-monitored blood glucose measurements as follows: upon rising, 1 h pre-prandial for all planned meals, 2 h post-prandial, and at bedtime. The daily calorie intake for diet therapy at our hospital was then calculated as follows: early gestation, 30 kcal × standard body weight (SBW; calculated as maternal height [m] × maternal height [m] × 22 kg) + 150 kcal; late gestation, 30 kcal × SBW + 350 kcal; overweight and obese, 30 kcal × SBW throughout pregnancy [7]. The nutritional status of the Diet GDM group at the Keio University Hospital is shown in Appendix A. If the patient had a fasting plasma glucose level (PG) ≥ 100 mg/dL or 2 h PG after meal ≥ 120 mg/dL, insulin therapy was initiated to improve hyperglycemia. The initial increase, subsequent decrease, and gestational weight gain (GWG) expected at 40 weeks of gestation were calculated as previously described [13,14].

Data are presented as the median (range) or number (percentage). Continuous data were compared between groups using the Mann–Whitney U test. Categorical variables were analyzed using the chi-square test or Fisher’s exact test. The trend for the number of abnormal values in the 75 g OGTT was analyzed using Cochran–Armitage trend analysis. Multiple logistic regression analyses were performed to determine the relative contributions of parameters (*p* < 0.1) to LGA in the study participants and Diet Early GDM. Statistical analyses were performed using JMP software (ver. 17, SAS Institute, Cary, NC, USA), and *p* < 0.05 was considered statistically significant.

## 3. Results

Based on the PSM adjusted for maternal age, parity, and pre-pregnancy BMI, we evaluated the maternal characteristics and perinatal outcomes between the Diet Early GDM (*n* = 309) and NGT (*n* = 309) groups (Table 1). The number of calorie intake categories for diet therapy in patients with Early GDM is shown in Appendix A. The calorie intake during early gestation was between 1400 and 2100 kcal/day, between 1400 and 2300 kcal/day during late gestation. The GWG expected at 40 gestational weeks in the Diet Early GDM group was significantly lower than that of the NGT group (*p* = 0.0086). In perinatal outcomes, the incidence of low birth weight (<2500 g) in the NGT group was significantly lower than in the early GDM group (*p* = 0.0019), and the Apgar score at 5 min in the NGT group was significantly lower than in the early GDM group (*p* = 0.0006). No significant differences were found in other maternal and perinatal outcomes between the two groups.

The comparison of characteristics and perinatal outcomes between LGA and non-LGA among 618 mothers is presented in Table 2. The GWG expected at 40 gestational weeks and gestational weeks at delivery in LGA were significantly higher than those in non-LGA (*p* = 0.0009 and 0.0014, respectively). However, the incidence of Diet GDM was similar between the two groups (*p* = 0.40). Multiple logistic regression analysis revealed that pre-pregnancy BMI and expected GWG at 40 weeks were significantly associated with LGA (Table 3).

Comparisons between LGA and non-LGA infants in the early Diet GDM group are shown in Appendix A. Pre-pregnancy BMI, expected GWG at 40 gestational weeks, and gestational weeks at delivery are significantly higher in the non-LGA group (*p* = 0.017, 0.023, and 0.046, respectively). However, no differences were found in random PG in the first trimester, 75 g OGTT values, initial increase, or subsequent decrease between the two groups. Multiple logistic regression analysis revealed that pre-pregnancy BMI and expected GWG at 40 weeks are significantly associated with LGA in Diet Early GDM (Appendix A).

## 4. Discussion

The perinatal outcomes in the Diet Early GDM group were no worse than those in the patients with NGT selected using PMS. In this study, a higher GWG expected at 40 gestational weeks and GWG at delivery were risk factors for LGA.

Based on our data, Early GDM did not adversely affect perinatal outcomes, although Late GDM was associated with adverse perinatal outcomes in a previous report [5]. The Japanese GDM diagnostic criteria adopted cutoff values from the International Association of Diabetes and Pregnancy Study Group (IADPSG). However, IADPSG cutoff values are suitable for Late GDM and are used to diagnose Early GDM in Japan [15]. Of the mothers with abnormal 75 g OGTT values in the first trimester, approximately half did not have abnormal 75 g OGTT values when they repeated the 75 g OGTT between 24 and 28 gestational weeks [6,16]. No differences were found in perinatal outcomes between mothers with Late abnormal 75 g OGTT and those with Late normal 75 g OGTT among Ealy abnormal 75 g OGTT who deferred treatment for GDM in the TOBOGM study [17]. Differences in the perinatal outcomes of diet therapy between early and late GDM might be affected by these criteria. Asians, including Japanese, exhibit impaired insulin secretion, which is an important mechanism for GDM development [18,19,20,21,22]. There are many underweight (pre-pregnancy BMI < 18.5 kg/m^2^) cases with GDM in Japan [23]. Therefore, differences in maternal metabolic characteristics between Japanese and Caucasian populations may have affected the results. We need to reconsider which mothers with Early GDM should be treated. Because maternal age at delivery, pre-pregnancy BMI, hemoglobin A1c in the first trimester, 1h PG, 2h PG, initial increase, subsequent decrease, and number of abnormal values in the 75 g OGTT were significantly higher in Early GDM treated with insulin therapy that was initiated before 24 gestational weeks than in Diet Early GDM. 

Some diet therapies for GDM have been reported. Dietary approaches to stop hypertension (DASH) are associated with decreased neonatal overgrowth and adverse perinatal outcomes [24]. The Mediterranean Diet reduced the development of GDM, hypertensive disorders of pregnancy, preeclampsia, preterm delivery, low birth weight (<2500 g), and fetal growth restriction [25]. Tsirou et al. reported that a very-low-calorie diet regimen (1600 kcal/day) and a low-calorie diet regimen (1800 kcal/day) achieved similar maternal, neonatal, and perinatal outcomes [26]. A systematic review and meta-analysis showed that diet therapy for GDM can improve maternal glycemic parameters and perinatal outcomes [27]. However, to date, no studies have confirmed the appropriate calorie intake for mothers with GDM. A Cochrane review reported that the evidence for diet therapy in GDM remains limited [28]. Recently, several randomized trials have been conducted. The aim of reducing hyperglycemia was the New Nordic Diet in women in the Gestational Onal Diabetes Mellitus (iNDIGO) study, a randomized parallel controlled trial in Sweden [29]. Manchester intermittent diet in GDM acceptability study (MIDDAS-GDM) is planned to investigate the safety, feasibility, and acceptability of an intermittent low-energy diet for patients with obesity [30]. In this study, maternal weight, not only during pregnancy but also before pregnancy, was associated with LGA in Diet Early GDM. According to preconception care, controlling maternal weight is also important for improving neonatal overgrowth in the Diet Early GDM. 

This study had some limitations. First, it was a retrospective cohort study. Therefore, when mothers with early GDM do not receive treatment to control PG, we do not know how the perinatal outcomes are impacted. However, to the best of our knowledge, no data exist comparing perinatal outcomes between dietary early GDM and NGT. Additionally, since we evaluated the perinatal outcomes between the two groups using PSM, the bias could be minimized, and the results might be useful and important for managing early GDM in the diet. Second, because our hospital is a university hospital, many high-risk mothers may have been included in the NGT group. Third, in our hospital, nutritionists educated every patient with Early GDM on calorie intake. However, we could not obtain detailed information about the diet of mothers with early GDM. Furthermore, although we analyzed the comparison of maternal and perinatal outcomes between Diet Early GDM and NGT, mothers with NGT did not receive any nutritional advice, and we did not receive any data about calorie intake from them. However, in the previous report, the mean Japanese pregnant women’s calorie intake was <1600 kcal [31]. Therefore, the calorie intake of mothers with NGT might be lower in Japanese than that found in our data (Appendix A). Mustafa et al. reported that low adherence in the diet recommendation group was associated with increased oral medication and insulin therapy compared to the high adherence group in mothers with GDM. Visiting a nutritionist is associated with decreased LGA infants [32]. Moreover, primiparity, no history of GDM, underweight, and smoking were significantly associated with lower dietary adherence in New Zealand [33]. Although most expectant women at our hospital are nulliparous and do not smoke, and adherence to diet therapy might be high, further research is required to investigate how mothers with Early GDM select and eat, using tools such as smartphone applications. Furthermore, although the American Diabetes Association recommends that metformin and glyburide should not be used as first-line agents and other oral medications to reduce maternal PG, long-term safety data are unavailable [2]. Oral medications are used to manage GDM in other countries. However, oral medications are not administered to pregnant women in Japan. Therefore, the association between oral medication and dietary therapy was not evaluated in this study.

## 5. Conclusions

Among patients with early GDM, those treated with diet therapy alone until delivery did not exhibit a worse prognosis than those treated with NGT. To prevent LGA, it may be important to control maternal body weight not only during pregnancy but also before conception.

## Figures and Tables

**Table 1 nutrients-16-01553-t001:** Characteristics and perinatal outcomes between diet therapy in early gestational diabetes group and normal glucose tolerance group.

		Diet Early GDM		NGT		*p*-Value
		(*n* = 309)		(*n* = 309)		
Maternal age at delivery	(years)	37	(24–51)	37	(24–48)	0.48
Nulliparity		180	(58%)	173	(56%)	0.63
Pre-pregnancy BMI	(kg/m^2^)	21.2	(16.0–36.3)	21.5	(15.5–35.9)	0.51
Maternal pre-pregnancy BMI category						0.39
Underweight (BMI < 18.5)		38	(12%)	37	(12%)	
Normal weight (18.5 ≤ BMI < 25.0)		213	(69%)	223	(72%)	
Overweight (25.0 ≤ BMI < 30.0)		51	(17%)	39	(13%)	
Obese (30 ≤ BMI)		7	(2%)	10	(3%)	
Gestational weight gain expected 40 weeks	(kg)	8.7	(−8.6, 21.3)	9.8	(−13.7, 24)	0.0086
Gestational weeks at delivery	(weeks)	38	(23–41)	38	(22–41)	0.36
Preterm delivery		39	(13%)	57	(18%)	0.059
Cesarean section delivery		149	(48%)	164	(53%)	0.26
Birthweight	(g)	2906	(438–4526)	2930	(257–4276)	0.79
Hypertensive disorder of pregnancy		11	(4%)	21	(7%)	0.10
Macrosomia (birthweight ≥ 4000 g)		1	(0%)	4	(1%)	0.37
Low birthweight (birthweight < 2500 g)		38	(12%)	68	(22%)	0.0019
Large for gestational age (birthweight ≥ 90 percentile)		35	(11%)	43	(14%)	0.40
Small for gestational age (birthweight < 10 percentile)		20	(6%)	28	(9%)	0.29
Apgar score 1 min		8	(1–10)	8	(0–10)	0.17
<6		9	(%)	36	(%)	<0.001
<8		47	(%)	81	(%)	0.001
Apgar score 5 min		9	(2–10)	9	(0–10)	0.0006
<6		2	(%)	13	(%)	0.0034
<8		8	(%)	36	(%)	<0.001

BMI, body mass index; GDM, gestational diabetes; NGT, normal glucose tolerance.

**Table 2 nutrients-16-01553-t002:** Characteristics and perinatal outcomes between large and non-large for gestational age groups in this study.

		LGA		Non-LGA		*p*-Value
		(*n* = 78)		(*n* = 540)		
Maternal age at delivery	(years)	37	(28–45)	36.5	(24–51)	0.39
Nulliparity		48	(62%)	305	(56%)	0.46
Pre-pregnancy BMI	(kg/m^2^)	21.7	(17.5–32.4)	21.3	(15.5–36.3)	0.37
Maternal pre-pregnancy BMI category						0.55
Underweight (BMI < 18.5)		9	(12%)	66	(12%)	
Normal weight (18.5 ≤ BMI < 25.0)		51	(65%)	385	(71%)	
Overweight (25.0 ≤ BMI < 30.0)		15	(19%)	75	(14%)	
Obese (30 ≤ BMI)		3	(4%)	14	(3%)	
Diet Early GDM		35	(45%)	274	(51%)	0.40
Random plasma glucose level during the first trimester	(mg/dL)	98	(82–143)	99	(60–169)	0.55
Gestational weight gain expected 40 weeks	(kg)	11.4	(1.8–24.0)	9	(−13.7–23.8)	0.0009
Gestational weeks at delivery	(weeks)	39	(24–40)	38	(22–41)	0.0014
Preterm delivery		8	(10%)	88	(16%)	0.19
Cesarean section delivery		31	(40%)	282	(52%)	0.04
Birthweight	(g)	3613	(897–4526)	2866	(257–3722)	<0.0001
Apgar score 1 min		8	(1–9)	8	(0–10)	0.78
Apgar score 5 min		9	(2–10)	9	(0–10)	0.84

LGA: large for gestational age; BMI, body mass index; GDM, gestational diabetes.

**Table 3 nutrients-16-01553-t003:** Associations of maternal and perinatal factors with large for gestational age.

Variable	Unadjusted OR (95%CI)		*p*-Value	Adjusted OR (95%CI)		*p*-Value
Gestational weight gain expected 40 weeks (+1 kg)	1.08	(1.03–1.14)	0.0019	1.09	(1.03–1.14)	0.0016
Gestational weeks at delivery (+1 week)	1.14	(1.02–1.30)	0.016	1.14	(1.02–1.30)	0.014

CI: confidence interval.

## Data Availability

The data presented in this study are available on request from the corresponding author. The data are not publicly available due to privacy.

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
