# Peer review of "Perinatal Outcomes of Diet Therapy in Gestational Diabetes Mellitus Diagnosed before 24 Gestational Weeks"

_nutrients, 2024, doi:10.3390/nu16111553_

Round 1

Reviewer 1 Report

Comments and Suggestions for Authors

This retrospective series is of interest. 

More detail should be presented about the screening method which allowed early detection of GDM. 

Insulin therapy was used - but it is vital to include data on how many of the cohort received insulin and at what gestational age it was started. 

Consideration is required of use of the term 'macrosomia' for a Japanese population. Is it really appropriate given the generally smaller body size compared with US or European women? 

Similarly, use of the terms ones and overweight need to be justified in this population. 

Author Response

This retrospective series is of interest. 

More detail should be presented about the screening method which allowed early detection of GDM. 

[Response]

Thank you for your insightful comment. Accordingly, we have added the sentence and Supplementary Figure. However, if you want us to describe the criteria in detail, we can add them to the Methods section.

[Method]

“The diagnostic criteria for GDM in this study are shown in Supplementary Figure S1 [7,8].” (p2, lines 69–70)

[Supplementary Materials]

“Figure S1: Diagnostic criteria for gestational diabetes mellitus.” (p6, line 223)

Insulin therapy was used - but it is vital to include data on how many of the cohort received insulin and at what gestational age it was started. 

[Response]

Thank you for your query. As you know, Diet GDM mothers were not treated by insulin therapy in this study. Therefore, we could not describe the timing starting insulin therapy. However, if Early GDM mothers need insulin therapy, they receive it as sson as possible.

Consideration is required of use of the term 'macrosomia' for a Japanese population. Is it really appropriate given the generally smaller body size compared with US or European women? 

Similarly, use of the terms ones and overweight need to be justified in this population. 

[Response]

Thank you for your comment. You may be right, but this definition remains the same in Japan.

Reviewer 2 Report

Comments and Suggestions for Authors

This paper reports a retrospective, (case-control) study  comparing perinatal outcomes in mother/newborn pairs exposed to diet treatment and in matched pairs, over a long period of time, in Japan.

Main criticisms

1.     Table 2 analyzed LGA vs AGA/SGA only in one group, then, table 3 reports a regression only in this group. This is not acceptable.

LGA babies in the population under study are 35 + 43. Therefore, to evaluate effects of diet treatment on LGA occurrence, the following steps are required: 1)  comparison in table 1 are OK as it is, then

2) the entire population of 618 women should be analyzed comparing LGA vs AGA-SGA (new Table 2),  then

3) variables of the 98 LGA babies which differ by p<0.10 should be utilized to perform a multiple  regression.  Authors could add diet or not as bivariate variable to finally look at diet effect even diet group did not show a significant different occurrence of LGA.

2.     Women receiving diet controls glucose levels 7 times/day and in case of anomaly they refer to doctor for treatment. Hence, they are managed more carefully than controls. Please discuss in detail effect of diet vs effects of different management.

Line 76:   Please, explicit diagnostic criteria for early GDM.

Line 89:     plasma glucose level (PG) <100 mg/dL or 2h-PG after meal <120 mg/dL”    It is perhaps > and > , I suppose.

Table 1: Apgar score is a categorical variable where clinical evaluation required to report data stratified by  categories (Apgar<8, or <6),  then analyze.

The point 1 is of paramount importance for me. Therefore, the paper should be rewritten according to results of the advised regression, otherwise rejected.

Why make efforts to build a matched control group if their data are not used in the final regression?

Comments on the Quality of English Language

The paper should has a major flaw and shoul be rewritten according to results of the advised regression, otherwise rejected.

Why make efforts to build a matched control group if their data are not used in the final regression?

Author Response

Main criticisms

  1. Table 2 analyzed LGA vs AGA/SGA only in one group, then, table 3 reports a regression only in this group. This is not acceptable.

LGA babies in the population under study are 35 + 43. Therefore, to evaluate effects of diet treatment on LGA occurrence, the following steps are required: 1)  comparison in table 1 are OK as it is, then

2) the entire population of 618 women should be analyzed comparing LGA vs AGA-SGA (new Table 2),  then

3) variables of the 98 LGA babies which differ by p<0.10 should be utilized to perform a multiple regression.  Authors could add diet or not as bivariate variable to finally look at diet effect even diet group did not show a significant different occurrence of LGA.

[Response]

Thank you for your valuable comment. We have analyzed the association between LGA and non-LGA among 618 mothers and created a new Table 2. However, as described in the Methods section, most mothers with NGT did not receive OGTT. Therefore, we have deleted the OGTT values from the new Table 2. Furthermore, since the incidence of Diet GDM was similar between the two groups, we could not analyze the association between Diet GDM and LGA using multiple logistic analysis. Is this correct? If our perception is incorrect, please tell us. We will analyze it again. However, we are concerned that you may not understand our argument. Specifically, we would like to argue that although Diet GDM does not have a worse prognosis than NGT, we should actively manage those Diet GDM cases who are at risk of developing LGA. We are concerned that the results that you have directed us to remove will not convey this assertion. Would you please allow us to post the data you have instructed us to delete, as well as the Supplementary Table? If you do not agree with our suggestion, we can delete these data.

[Methods]

“Multiple logistic regression analyses were performed to determine the relative contributions of parameters (p <0.1) to LGA in this study participants and Diet Early GDM.” (p3, lines 100–101)

[Results]

“The comparison of characteristics and perinatal outcomes between LGA and non-LGA among 618 mothers is presented in Table 2. GWG expected at 40 gestational weeks and gestational weeks at delivery in LGA were significantly higher than those in non-LGA (p=0.0009 and 0.0014, respectively). However, the incidence of Diet GDM was similar between the two groups (p=0.40). Multiple logistic regression analysis revealed that pre-pregnancy BMI and expected GWG at 40 weeks were significantly associated with LGA (Table 3).” (p4, lines 123–129)

[Discussion]

“In this study, higher GWG expected at 40 gestational weeks and GWG were risk factors for LGA.” (p5, lines 145–146)

  1. Women receiving diet controls glucose levels 7 times/day and in case of anomaly they refer to doctor for treatment. Hence, they are managed more carefully than controls. Please discuss in detail effect of diet vs effects of different management.

 [Response]

Thank you for your important comment. Accordingly, we have added the following sentence and reference.

[Discussion]

“Furthermore, although we analyzed the comparison of maternal and perinatal outcomes between Diet Early GDM and NGT, mothers with NGT did not receive any nutritional advice, and we did not receive any data about calorie intake from them. However, in the previous report, mean Japanese pregnant women’s calorie intake was <1,600 kcal [31]. Therefore, the calorie intake of mothers with NGT might be lower in Japanese than that found in our data (Supplementary Figure S2).” (p6, lines 196–202)

[Reference]

  1. Kubota, K.; Itoh, H.; Tasaka, M.; Naito, H.; Fukuoka, Y.; Muramatsu Kato, K.; Kohmura, Y.K.; Sugihara, K.; Kanayama, N.; Hamamatsu Birth Cohort Study, T. Changes of maternal dietary intake, bodyweight and fetal growth throughout pregnancy in pregnant Japanese women. J. Obstet. Gynaecol. Res. 2013, 39, 1383–1390, doi:10.1111/jog.12070.

Line 76:   Please, explicit diagnostic criteria for early GDM.

[Response]

Thank you for your insightful comment. Accordingly, we have added the sentence and Supplementary Figure. However, if you want us to describe the criteria in detail, we can add them to the Methods section.

[Method]

“The diagnostic criteria for GDM in this study are shown in Supplementary Figure S1 [7,8].” (p2, lines 69–70)

[Supplementary Materials]

“Figure S1: Diagnostic criteria for gestational diabetes mellitus.” (p6, line 223)

Line 89:     “plasma glucose level (PG) <100 mg/dL or 2h-PG after meal <120 mg/dL”    It is perhaps > and > , I suppose.

[Response]

Thank you for your comment. We have revised the sentence.

[Methods]

“If the patient had a fasting plasma glucose level (PG) ≥100 mg/dL or 2h-PG after meal ≥120 mg/dL, insulin therapy was initiated to improve hyperglycemia.” (p2, lines 91–93)

Table 1: Apgar score is a categorical variable where clinical evaluation required to report data stratified by categories (Apgar<8, or <6),  then analyze.

 [Response]

Thank you for your valuable comment. Accordingly, we have added the data to Table 1.

Reviewer 3 Report

Comments and Suggestions for Authors

Kasuga et al studied perinatal outcome of diet therapy for the management of early gestational diabetes mellitus. The paper is well structured, the results are interesting for the readers. I have only few suggestions: 

in the introduction a definition of gestational diabetes mellitus should be added. I think that the diagnosic criteria, now reported in the methods, could be anticiped in the introduction. 

In the introduction the consequences for the neonate should be explained.

Author Response

Kasuga et al studied perinatal outcome of diet therapy for the management of early gestational diabetes mellitus. The paper is well structured, the results are interesting for the readers. I have only few suggestions: 

in the introduction a definition of gestational diabetes mellitus should be added. I think that the diagnosic criteria, now reported in the methods, could be anticiped in the introduction. 

[Response]

Thank you for your comment. The diagnostic criteria have been moved from the Methods section to the Introduction section.

Introduction:

“Early GDM was diagnosed using the criteria published by the Japanese Society of Obstetrics and Gynecology and our hospital’s criteria described in our previous reports [6,7].” (p2, lines 51–53)

In the introduction the consequences for the neonate should be explained.

[Response]

Thank you for your editorial comment. Accordingly, we have added the following sentence and reference:

[Introduction]

“Gestational diabetes mellitus (GDM) is a high-incidence perinatal complication, and neonates born to mothers with GDM are at higher risk of large for gestational age (LGA: birthweight ≥90%percentile), macrosomia (birthweight ≥4000 g), neonatal hypoglycemia, and jaundice [1].” (p1, lines 27–30).

[Reference]

  1. ACOG Practice Bulletin No. 190: Gestational diabetes mellitus. Gynecol. 2018, 131, e49–e64, doi:10.1097/AOG.0000000000002501.